# Vutiglabridin Alleviates Cellular Senescence with Metabolic Regulation and Circadian Clock in Human Dermal Fibroblasts

**DOI:** 10.3390/antiox13010109

**Published:** 2024-01-16

**Authors:** Jin-Woong Heo, Hye-Eun Lee, Jimin Lee, Leo Sungwong Choi, Jaejin Shin, Ji-Young Mun, Hyung-Soon Park, Sang-Chul Park, Chang-Hoon Nam

**Affiliations:** 1School of Undergraduate Studies, Daegu Gyeongbuk Institute of Science and Technology, College of Transdisciplinary Studies, Daegu 42988, Republic of Korea; hjw001107@dgist.ac.kr (J.-W.H.); jimin6440@dgist.ac.kr (J.L.); 2Aging and Immunity Laboratory, Department of New Biology, Daegu Gyeongbuk Institute of Science and Technology, Daegu 42988, Republic of Korea; 3School of Medicine, Kyungpook National University, Daegu 41566, Republic of Korea; heni91@kbri.re.kr; 4Neural Circuit Research Group, Korea Brain Research Institute, Daegu 41062, Republic of Korea; jymun@kbri.re.kr; 5Glaceum Incorporation, Research Department, Suwon 16675, Republic of Korea; leochoi@glaceum.com (L.S.C.); jaejin@glaceum.com (J.S.); hspark@glaceum.com (H.-S.P.); 6Future Life and Society Research Center, Advanced Institute of Aging Science, Chonnam National University, Gwangju 61186, Republic of Korea; scpark@snu.ac.kr

**Keywords:** human dermal fibroblasts, cellular senescence, circadian clocks, metabolism, mitochondrial homeostasis

## Abstract

The process of cellular senescence, which is characterized by stable cell cycle arrest, is strongly associated with dysfunctional cellular metabolism and circadian rhythmicity, both of which are reported to result from and also be causal to cellular senescence. As a result, modifying any of them—senescence, metabolism, or the circadian clock—may affect all three simultaneously. Obesity accelerates aging by disrupting the homeostasis of reactive oxygen species (ROS) via an increased mitochondrial burden of fatty acid oxidation. As a result, if senescence, metabolism, and circadian rhythm are all linked, anti-obesity treatments may improve metabolic regulation while also alleviating senescence and circadian rhythm. Vutiglabridin is a small molecule in clinical trials that improves obesity by enhancing mitochondrial function. We found that chronic treatment of senescent primary human dermal fibroblasts (HDFs) with vutiglabridin alleviates all investigated markers of cellular senescence (SA-β-gal, CDKN1A, CDKN2A) and dysfunctional cellular circadian rhythm (BMAL1) while remarkably preventing the alterations of mitochondrial function and structure that occur during the process of cellular senescence. Our results demonstrate the significant senescence-alleviating effects of vutiglabridin, specifically with the restoration of cellular circadian rhythmicity and metabolic regulation. These data support the potential development of vutiglabridin against aging-associated diseases and corroborate the intricate link between cellular senescence, metabolism, and the circadian clock.

## 1. Introduction

Cellular senescence, characterized by stable cell cycle arrest, has been identified as a major contributor to aging and age-related diseases [1], and has thus emerged as a promising target for the search for potential therapeutics [2]. Senescence is thought to be caused by various exogenous and endogenous factors, such as telomere dysfunction, DNA damage, and organelle stress [3,4]. The most common senescent cell marker is Senescence-associated β galactosidase (SA-β-gal) [5], and p21 and p16 are two cyclin-dependent kinase inhibitors that are essential mediators of senescence-associated cell cycle arrest [6]. Furthermore, when cells are cultured for a long time over several passages, various cellular senescence phenomena are observed, and HDFs have been commonly used for the investigation of human-relevant cellular senescence [7].

The process of cellular senescence causes and is intricately linked to alterations in cellular metabolism, as demonstrated by the disruption of reactive oxygen species (ROS) homeostasis and mitochondrial dysfunction [8]. Mitochondria are the primary site of cellular metabolic regulation, synthesizing ATP using various coenzymes such as NAD^+^ and generating ROS as an unavoidable by-product [9]. ROS and mitochondria are also strongly associated with alterations in the levels of sirtuins, which are critical regulators of ROS, mitochondria, and aging, and AMP-activated protein kinase (AMPK), which is a master regulator of metabolism [10,11].

These metabolic changes induced by the process of cellular senescence also cause changes in circadian rhythmicity and are also consequences of changes in circadian rhythmicity [12]. Cellular senescence is known to trigger alterations in the circadian clock with prolonged and delayed phases [13], whereas unaltered and consistent circadian rhythm is generally associated with a young and healthy lifestyle [14]. Excessive ROS reduces the activity of REV-ERB via oxidation, thereby abnormally increasing the transcription of key clock genes, including BMAL1 [15]. Bmal1, which regulates the circadian clock, in return influences ROS production by inducing morphological changes in mitochondria [16]. In addition, the redox state of total NAD influences the circadian rhythm by affecting the DNA binding potential of the Bmal1:NPAS2 complex [12]. Furthermore, by increasing the AMP/ATP ratio, AMPK influences the circadian clock in senescent cells [12]. Overall, changes in cellular senescence and the circadian clock have a complex relationship with cellular metabolism dysfunction, though the precise mechanism linking them is unknown.

Previous studies provide evidence that the molecular components of the circadian clock may be essential regulators of metabolic changes at the cellular level, including redox homeostasis. Therefore, the maintenance of cellular and physiological circadian rhythms is generally reported to be crucial for maintaining metabolic health [17]. Metabolic processes in the liver are regulated by circadian gene oscillations [17]. In detail, high ROS loads were shown in the cells isolated from Bmal1−/− mice, which indicates that Bmal1 is involved in regulating oxidative stress in various tissues [18]. ROS production and the repairing system of the consequent DNA damages are adjusted periodically to minimize oxidative damage to the genome [19]. The metabolic changes generated through this process act as a feedback mechanism that again causes modulation of the cellular circadian rhythmicity.

Obesity exemplifies the imbalance in ROS homeostasis, mitochondrial dysfunction, and cellular senescence [20,21]. Obesity is strongly associated with excessive ROS levels in various cell types, which leads to cell damage and disease associated with aging [22]. The mice that developed exaggerated obesity displayed impaired spontaneous activity, mitochondrial dysfunction, and increased mitochondrial ROS production in skeletal muscle [23]. In human fibroblasts subjected to chronic excess ROS, mitochondrial DNA (mtDNA) was severely damaged as it is proximal to the electron transport chain (ETC), where ROS is being formed, and is not protected by chromatin and histone-like nuclear DNA [24]. Damaged mtDNA is continually multiplied in the cell and causes dysfunctional mitochondria formation with reduced energy production efficiency [25]. Obesity was reported to impair mitochondrial biogenesis [26], decrease mitochondrial content, reduce mtDNA stability, and impair mitochondrial fusion in skeletal muscle [27].

Glabridin is an anti-obesity compound with an isoflavan backbone and can be obtained naturally from the root extract of licorice [28]. Glabridin reduced lipogenesis via activating the signalling pathway of AMPK and contributed to reducing adipose tissue mass and adipocyte hypertrophy in high-fat diet(HFD)-induced obese mice [29]. However, glabridin was not readily developed as a drug due to its low chemical stability, which is greatly affected by temperature, illumination, humidity, and pH [30]. Therefore, vutiglabridin was developed by chemically modifying glabridin to improve its chemical stability and low bioavailability [30] and was found to supersede glabridin in the weight-reducing efficacy in HFD-induced obese mice [30]. Vutiglabridin was mainly found to regulate metabolic gene expression in the body’s major organs—liver, muscle, hypothalamus, and adipose tissue—and specifically increased AMPK signalling [31]. A recent study discovered that vutiglabridin improves mitochondrial function by binding to and activating paraoxonase-2 [31], a mitochondrial inner-membrane protein. It is unknown whether vutiglabridin has any effects on cellular senescence, metabolic regulation, or circadian rhythm disruption.

In this study, we hypothesized that vutiglabridin could ameliorate mitochondrial dysfunction and the cellular metabolism in senescent-processing cells and thereby restore the circadian phenotype of cellular senescence. To substantiate this hypothesis, we chronically applied vutiglabridin in the HDFs cultured over a long period to induce senescence. For the first time, we show that chronic vutiglabridin treatment slows the progression of cellular senescence, mitochondrial dysfunction, and the senescent circadian clock, demonstrating the strong link between senescence, metabolism, and circadian rhythmicity and providing support for the potential development of vutiglabridin against age-related diseases.

## 2. Materials and Methods

### 2.1. Cell Culture

Normal neonatal HDFs (PCS-201-010, American type culture collection) were cultured and serially passaged by Dr. Young-Sam Lee. Passage 10 and passage 30 HDF groups were kindly provided for this research.

Cells were grown in Dulbecco’s Modified Eagle medium (Welgene, Gyeongsan, Republic of Korea) supplemented with 10% fetal bovine serum (Gibco, Billings, MT, USA) and Antibiotics (Welgene, Gyeongsan, Republic of Korea) at 37 °C, 5% CO_2_.

### 2.2. Cell Growth Assay

HDFs were cultured in triplicates in 6 well plates. The seeding density was 105 cells/well. After passage 45, vutiglabridin-treated HDFs were seeded 8 × 10^4^ cells/well, and DMSO-treated HDFs were seeded 6 × 10^4^ cells/well. After cell adhesion, 20 μM of vutiglabridin (provided by Glaceum Inc., Suwon-si, Republic of Korea) was treated in vutiglabridin-treated HDF groups. The concentration used in this study was 20 μM, which has been optimized as the human-relevant dose. Furthermore, the same volume of DMSO was treated in DMSO-treated groups. If the cells reached semi-confluency, cells were treated with trypsin, counted, and seeded.

Population doubling level (PDL) was calculated by using the following formula: PDL = log_2_ (harvested cell number/seeded cell number).

The cumulative PDL (cPDL) of passage was calculated by accumulating all of the previous PDL.

Doubling time was calculated using the following formula:Doubling time = (harvested day − seeded day)/population doubling level

In this study, ‘senescent cells’ were defined as cells that had a doubling time of more than 7 days. While this does not represent permanent cell cycle arrest, we used this standard in order to accurately perform circadian clock experiments, which mandated that fully senescent cells could not be used due to their low resilience [32], and yet significantly observe markers of senescence-progressing cells.

### 2.3. Senescence Associated-β-Galactosidase (SA-β-gal) Assay

The SA-β-gal assay was performed with a senescence beta-galactosidase staining kit (Cell Signaling Technology, Danvers, MA, USA). β-galactosidase staining solution and the fixative solution were arranged with the manufacturer’s instructions. After treating 1 mL of 1× fixative solution in HDF seeded in a 6-well plate, they were washed twice with DPBS. Next, HDFs were incubated in a 37 °C dry incubator with 1 mL of X-gal staining solution. The SA-β-gal positive quantification was performed by manually counting 200 cells for blue staining in each sample under a bright-field microscope [33].

### 2.4. RNA Extraction and cDNA Synthesis

Total RNA extraction from HDFs was performed with an RNeasy mini kit (Qiagen, Hilden, Germany). The extracted total RNA was reverse-transcribed with TOPscriptTM Reverse transcriptase (Enzynomics, Daejeon, Republic of Korea) and Oligo dT primers.

### 2.5. Quantitative Real-Time PCR

Quantitative PCR was performed with TOPreal^TM^ qPCR 2× premix (Enzynomics, Daejeon, Republic of Korea) and CFX96 Touch Deep Well Real-Time PCR Detection System (Bio-Rad, Hercules, CA, USA).

The following condition was used: 95 °C for 15 min, 44 cycles at 95 °C for 10 s, 60 °C for 15 s, and 72 °C for 30 s.

The primer sequence was as follows:

*P21 (CDKN1A)* forward: 5′-CCGCCCCCTCCTCTAGCTGT-3′

*P21 (CDKN1A)* reverse: 5′-CCCCCATCATATACCCCTAACACA-3′

accession number: AF497972.1

*P16 (CDKN2A)* forward: 5′-CCCCGATTGAAAGAACCAGAGA-3′

*P16 (CDKN2A)* reverse: 5′-ACGGTAGTGGGGGAAGGCATAT-3′

accession number: AB060808.1

*GAPDH1* forward: 5′-GAAGGTGAAGGTCGGAGT-3′

*GAPDH1* reverse: 5′-GAAGATGGTGATGGGATTTC-3′

accession number: AA202964

### 2.6. Extracellular Flux Assays

The glycolysis rate and the oxygen consumption rate (OCR) in HDFs were measured with a Seahorse XFe24 analyzer (Agilent: Seahorse XFe24 analyzer, Santa Clara, CA, USA) and Wave Software (Agilent: wave controller, https://www.agilent.com/en/product/cell-analysis/real-time-cell-metabolic-analysis/xf-software/seahorse-wave-desktop-software-740897, accessed on 5 March 2022). HDFs were seeded the previous day to reach 95% confluency before measuring. The glycolytic stress test measured the extracellular acidification rate (ECAR) following the manufacturer’s instructions. The OCR was measured using the Mito Stress test. DMEM media supplemented with 2 mM of L-glutamine (Thermofisher, Waltham, MA, USA) was used for the glycolytic stress test. A total of 10 mM of glucose (Agilent), 1 μM of oligomycin (Agilent), and 50 mM of 2-Deoxy-D-glucose (2-DG) (Agilent) were injected during the assay. For the MitoStress test, DMEM media supplemented with 10 mM of glucose, 1 mM of sodium pyruvate (Agilent), and 2 mM of L-glutamine were used. In total, 1.5 μM of oligomycin, 2 μM of FCCP (Agilent), and 0.5 μM of rotenone AA (Agilent) were injected during the assay.

### 2.7. Confocal Microscopy and Electron Microscopy

Cells in each group were grown in 35 mm glass-bottomed culture dishes (NEST Biotechnology Co., Wuxi, China, 801001) to 50–60% confluency. The next day, cells were incubated with 200 nM MitoTracker (Invitrogen, Carlsbad, CA, USA, M7514) for 20 min, then washed with PBS and examined under a confocal microscope (Olympus Corporation, Tokyo, Japan) to investigate structural changes of mitochondria.

Ultrastructural analysis of mitochondria was carried out by transmission electron microscopy (TEM). The cells on the coverslip were fixed with 2.5% glutaraldehyde-mixed 2% paraformaldehyde solution for 1 h, followed by post-fixation in 2% osmium tetroxide for 1 h at 4 °C. The fixed cells were dehydrated with a graded ethanol series and then embedded into an epoxy medium (EMS, Hatfield, PA, USA). Embedded samples were sectioned (60 nm) with an ultra-microtome (Leica Microsystems, Wetzlar, Germany), and the sections were then viewed on a Tecnai 20 TEM (Thermo Fisher Scientific, Waltham, MA, USA) at 120 kV. Then they were double-stained with UranyLess (EMS, 22409) for 2 min and 3% lead citrate (EMS, 22410) for 1 min. Images were captured with a US1000X-P camera 200 (Gatan, Pleasanton, CA, USA).

### 2.8. Western Blot

Resuspending HDF cell pellets were prepared for protein lysates with SDS sample buffer (50 mM Tris-HCl 2% SDS, 0.1% Bromophenol blue, 10% glycerol). Protein quantification was performed with an RC DC protein assay kit (Bio-Rad, Hercules, CA, USA). 15 μg of protein in each sample was separated in 10% Tris-glycine gel and transferred to a nitrocellulose membrane. Membranes were blocked with TBST buffer (20 mM Tris, 150 mM NaCl, 0.5% Tween-20) supplemented with 5% BSA and incubated with primary antibody. After washing with TBST, membranes were incubated with HRP-conjugated secondary antibodies. Similarly, after washing with TBST, an enhanced chemiluminescence solution (Thermo Scientific, Waltham, MA, USA) was added for detection. The primary antibodies are: Rabbit anti-AMPK (2534; Cell Signaling Technology, Danvers, MA, USA, 1:1000 dilution), Rabbit anti-pAMPK (2535; Cell Signaling Technology, Danvers, MA, USA, 1:1000 dilution), Rabbit anti-β actin (8457; Cell Signaling Technology, Danvers, MA, USA, 1:1000 dilution). The secondary antibodies are: Peroxidase AffiniPure Donkey Anti-Rabbit IgG (711-035-152; Jackson Immunoresearch, West Grove, PA, USA, 1:5000 dilution).

### 2.9. NAD^+^ and NADH Measurement

NAD^+^ and NADH measurements were performed with NAD^+^/NADH quantitation colorimetric kit (Biovision, Milpitas, CA, USA), following the manufacturer’s instructions. After pelleting and washing, the 2 × 10^5^ HDFs with DPBS, NAD^+^, and NADH were extracted. Extracted NAD^+^, NADH solution was filtered by a 10 kDa molecular cut-off filter (Biovision, Milpitas, CA, USA). NAD^+^ was degraded by heating at 60 °C for 30 min to measure the NADH. Next, the NAD cycling enzyme and NADH developer were added and incubated for 1 h at room temperature. Absorbance at 450 nm was measured. NAD^+^ concentration was calculated by subtracting NADH concentration from total NAD^+^ and NADH concentration.

### 2.10. Lentivirus Production

To investigate the level of BMAL1 gene expression in HDFs, a lentivirus vector expressing luciferase via the BMAL1 promoter was inserted into HDFs to generate a stable cell line. Dr. Steven A. Brown kindly provided a pLV6-BMAL1-Luc vector [34]. HEK293T cells were seeded in a 10 cm plate with 70~80% confluency. Transfection proceeded with the following protocol.

Three hours before the transfection media were removed, 10 mL of fresh media was added. In two 1.5 mL tubes, 625 μL of Opti-MEM (Gibco) was added. A total of 25 μL of lipofectamine 3000 (Invitrogen, Waltham, MA, USA) was added to one tube, and 9 μg of psPAX2 (Addgene, Watertown, MA, USA), 6 μg of psPAX2 (Addgene), 15 μg of pLV6-BMAL1-Luc vector (Addgene), and 25 μg of P3000 reagent were added. Tubes were incubated for 3 min at room temperature. Two tubes were mixed and incubated for 15 min at room temperature. A total of 4 mL of media was removed from the cell culture plate, and the mixture was added dropwise to the cell culture media. After 24 h incubation in the 37 °C, 5% CO_2_ incubator, media were changed with 12 mL of fresh media, and viral supernatant was collected; this process was repeated twice to gather a total of 36 mL of viral supernatant. Viral supernatant stored at −20 °C was thawed at 4 °C, centrifuged at 500× *g*, 5 min to remove the cell debris, filtered through the 0.45 μm filter, centrifuged at 80,000× *g*, 4 °C, for 2 h, and the lentivirus particles were collected.

### 2.11. BMAL1Luc HDFs Stable Cell Line Production

Lentivirus transduction was performed on HDFs when cells reached 30% confluency in a 10 cm dish. The culture medium was replaced with a lentivirus viral medium supplemented with 8 μg/mL of polybrene (Sigma, St. Louis, MI, USA). Eight hours later, cells were washed with DPBS and replaced with fresh media. The cells were cultured for an additional 3 days with changing culture media every day.

The selection for virus-infected HDFs was performed with 4 μg/mL blasticidin (Invivogen, San Diego, CA, USA) selection media. The selection proceeded for 6 days by replacing the blasticidin selection media every 2 days. Afterward, measurements were performed in a real-time luminometer (Atto: AB-2550 Kronos Dio, Manaus, Brazil), or the cells were banked.

### 2.12. Circadian Clock Synchronization and Real-Time Luciferase Monitoring Assay

The stable BMAL1Luc HDFs were seeded in a 35 mm dish. After adhesion, culture media were replaced with 2 mL of DMEM media supplemented with 200 nM dexamethasone (Sigma, St. Louis, MI, USA). Cells were incubated for 2 h at 37 °C with 5% CO_2_ to synchronize the cellular circadian clock. After synchronization, HDFs were washed with fresh media. The cultured media were replaced with 2 mL of DMEM media supplemented with 200 μM Beetle luciferin (Promega, Madison, WI, USA). BMAL1Luc HDFs’ bioluminescence was measured continuously with a Kronos Dio real-time luminometer (Atto: AB-2550 Kronos Dio). After 5 days, the raw data of bioluminescence was collected. Detrended data were collected from the Kronos Dio software to observe the circadian gene oscillation precisely.

### 2.13. Cosinor Analysis

Detrended bioluminescence data collected from a real-time luciferase monitoring assay were used for cosinor analysis, performed with cosinor software created by Dr. R. Refinetti (https://www.circadian.org/softwar.html, accessed on 7 November 2021).

## 3. Results

### 3.1. Vutiglabridin Treatment Alleviates Replicative Senescence of HDFs

The HDFs cultured with vutiglabridin displayed a significantly increased proliferation rate compared to the HDFs cultured with vehicle (DMSO)-containing media, as reflected by the number of doubling times and cumulative population doubling level (cPDL) occurring during the experiment. HDFs treated with 20 μM of vutiglabridin took 77 days until the doubling time reached 7 days, whereas HDFs treated with vehicle only took 58 days (Figure 1a). The curve relationship between cPDL and culture duration demonstrates a relatively linear decreasing PDL rate with time progression. Starting from day 24, HDFs cultured with vehicle began to have a decreasing rate of cPDL compared to HDFs treated with vutiglabridin. When the duration of cell culture reached day 84, the gap between the two groups widened significantly, indicating that the proliferative potential of HDFs cultured with vehicle decreased more quickly compared to HDFs treated with vutiglabridin (Figure 1b: Vuti:0.10 PD/day at day 82 vs. DMSO:0.07 PD/day at day 82; PD: population doubling).

For consequent experiments, senescent cells were defined as cells with a doubling time over 7 days for this study, where the cells were resilient for experimentation on circadian clock markers yet were surmised to have sufficient senescence characteristics. The time point when the doubling time of the DMSO-treated HDFs group exceeded 7 days was determined as the exact point of senescence, which was 58 days after culture. At this time point, the two HDF groups were defined as the ‘Vutiglabridin’ group and the ‘Old control’ group.

Representative pictures of the Vutiglabridin groups and the DMSO groups are shown in Figure 1c. Quantifying SA-β-gal positive cells was performed by counting 200 HDFs in each group as SA-β-gal positive or negative. Continuous treatment with vutiglabridin enhanced the proliferation of HDFs (Figure 1a,b). In addition, it showed a decreased ratio of SA-β-gal positive cells compared to treatments with DMSO (Figure 1d: 55.83% in vutiglabridin-treated vs. 67.50% in DMSO-treated, SA-β-gal positive cells).

The expression of cell cycle inhibitors p16 and p21 was measured to examine whether continuous treatment with vutiglabridin alleviates the process of replicative cellular senescence. The gene expression of P16 (CDKN2A) and P21 (CDKN1A) was measured in HDFs of passage 14 (Young control), passage 32 HDFs before vutiglabridin treatment (0 days), and HDFs treated with 20 μM of vutiglabridin for 58 days (Vutiglabridin day 58), and HDFs cultured for 58 days with DMSO (DMSO day 58).

Compared to the Young control, HDFs cultured with DMSO for 58 days showed P16 and P21 expression levels of 2.24 and 4.52-fold, respectively (Figure 1e, *p* < 0.001). In contrast, HDFs cultured with vutiglabridin for the same period showed P16 and P21 gene expression levels of 1.69- and 3.31-fold, respectively (Figure 1e, *p* < 0.001, *p* < 0.01), which were significantly lower than that of the DMSO group.

The lower ratio of SA-β-gal positive cells and the lower relative expression of P16 and P21 genes in the vutiglabridin-treated group show that chronic treatment with vutiglabridin alleviates the progression to cellular senescence in high-passage HDFs.

### 3.2. Long-Term Culture with Vutiglabridin Reduces Mitochondrial Dysfunction in High-Passage HDFs

The effect of vutiglabridin on mitochondrial dysfunction induced by the senescence process was evaluated, as this is the primary activity of vutiglabridin. The glycolysis stress test measured the ECAR after adding 10 mM of glucose to present the glycolysis rate (Figure 2a). The increased ECAR after 1 μM oligomycin treatment was defined as glycolytic capacity. The glycolysis rate and glycolytic capacity of the senescent HDF groups (Old control) were higher than those of the Young control groups (Young control) (Figure 2b, *p* < 0.001). The glycolysis level in the Old control group increased 4.08 times compared to the Young control group (Figure 2b). The vutiglabridin-treated HDF groups, on the other hand, demonstrated a lower glycolysis rate than the Old control group (Figure 2b, *p* < 0.001). The glycolysis level in the Vutiglabridin group was 0.37 times lower than in the Old control group (Figure 2b).

The Mito Stress test with each HDF group measures critical mitochondrial function parameters by directly measuring cell OCR. The oxygen consumption level in the Old control group was 1.41 times higher than in the Young control group (Figure 2d). However, in the Vutiglabridin group, oxygen consumption was reduced by 0.95 times, which was comparable to that of the Young control (Figure 2d).

Higher OCR and glycolysis rates in the Old control group are suggestive of mitochondrial dysfunction due to cellular senescence. Senescent cells have mitochondrial dysfunction and increased glycolysis ratio to meet their energy demands [37]. In addition, as mitophagy—the clearance of damaged mitochondria—decreases, the turnover rate of mitochondria is reduced, and mitochondria with reduced function are accumulated in the cytoplasm. As the number of dysfunctional mitochondria increases, the cellular OCR is reported to increase [38,39]. Treatment with vutiglabridin notably prevented such mitochondrial dysfunction in HDFs.

The mitochondrial structure as a marker of mitochondrial health in the Young and Old control group was investigated through confocal light microscopy and transmission electron microscopy (TEM). The abnormal mitochondrial structure, such as a fragmented network (Figure 2e), was observed in the Old control group. The confocal microscopy clearly showed the effect of vutiglabridin administration on alleviating such changes in mitochondrial structure. Confocal microscopy images from the Vutiglabridin group showed that the mitochondrial network structure is comparatively well maintained as in the Young control group (Figure 2e).

For its quantification, the mitochondria morphology was classified as individuals and branches. Rods, punctate, and round-shaped morphology were counted as individual mitochondria, but mitochondria with the network were categorized as branches. We investigated the number of individual mitochondria and branched mitochondria through confocal image analysis of each group (Figure 2f), which was measured by a junction pixel using the image J macro tool [40]. As a result, we identified significant increases in the number of individuals (average 48.3 vs. 91.1) and reduced branches (average 143.3 vs. 19.2) in Old HDFs compared to Young HDFs. In addition, branch lengths were decreased by 0.93 times in Old HDFs. However, treatment with vutiglabridin showed decreased individual mitochondria (average 152.6 vs. 91.1) and reduced short-branched mitochondria (average 19.2 vs. 145.7) in the Old HDFs. Lastly, TEM imaging of the mitochondrial ultrastructure revealed that the vutiglabridin treatment visibly reduced cristae disruption, mitochondrial swelling, and increased autophagic vesicles caused by long-term cell culture (Figure 2g).

Oxidative stress causes polyunsaturated fatty acid (PUFA) peroxidation and activates 4-hydroxy-2E-hexenal (4-HNE) [41]. 4-HNE is a biomarker of oxidative stress and a key mediator between oxidative stress and cellular senescence [42]. Senescent cells express more 4-HNE than young cells due to increased ROS levels [43]. The level of 4-HNE was measured to investigate ROS production in HDFs. The Old control group had higher expression than the Young control group, while the Vutiglabridin group had lower expression than the Old control group (Figure 2h). Overall, these data show that vutiglabridin indeed alleviates mitochondrial dysfunction and excess ROS generation in senescence-progressing HDFs.

### 3.3. Long-Term Culture with Vutiglabridin Modulates the Expression of the Metabolic Regulatory Protein in High-Passage HDFs

In order to assess the effects of vutiglabridin on metabolic regulation, the level of its key mediator AMPK was measured, and the AMPK phosphorylation level was measured by western blot, wherein its increase indicates activation of the AMPK signalling pathway. The Vutiglabridin group showed higher expression of pAMPKα than the Old control group (Figure 3a). pAMPKα/AMPKα value in the Vutiglabridin group was 2.01 times higher than the Old control group (Figure 3b). This result was consistent with the increased AMPK activation in mice treated with vutiglabridin [30]. This is further consistent with the anti-senescent effect of vutiglabridin, as the activation of AMPK inhibits mTORC1 signalling, slowing aging and prolonging lifespan [44,45].

Given that vutiglabridin activates AMPK signalling and modulates mitochondrial changes, the essential coenzyme NAD^+^ level and NAD^+^/NADH ratios were also investigated. The NAD^+^ levels and NAD^+^/NADH ratios of each HDF group were comparatively analyzed. The concentration of NAD^+^ was decreased by 0.39 times in the Old control group (Figure 3c), and the concentration of NADH was increased by 1.18 times in the Old control group (Figure 3d) compared to the Young control group. Compared to the Old control group, the Vutiglabridin group showed an increase of 1.76 times the NAD^+^ level and a decrease of 1.18 times of NADH level (Figure 3c, *p* < 0.01; Figure 3d, *p* < 0.05). Consequently, the Vutiglabridin group showed a 2.09 times higher NAD^+^/NADH ratio than the Old control group (Figure 3e, *p* < 0.01), overall showing a more remarkable similarity to the Young control group.

### 3.4. Long-Term Culture with Vutiglabridin Alleviates the Senescent Phenotype of the Circadian Clock in High-Passage HDFs

To investigate whether vutiglabridin as a metabolic regulator and senescence-alleviating compound also affects the cellular circadian clock, HDFs were treated with 20 μM of vutiglabridin or the same volume of DMSO and cultured for 58 days, and expression of the BMAL1-drive luciferase reporter was measured as shown in Figure 4. To investigate the BMAL1 gene expression pattern of HDFs, cells were infected with lentivirus harboring the BMAL1-Luc vector. After circadian clock synchronization with 200 nM of Dexamethasone for 2 h, the culture media were replaced with DMEM media supplemented with 200 μM of luciferin. Then the expression of the BMAL1-drive luciferase reporter was observed on a real-time luminometer for 5 days.

Samples were normalized by cell counting after a real-time luciferase monitoring assay to calculate the amplitude of the BMAL1 promoter expression pattern. The average BMAL1 promoter expression level was decreased in the Old control group. In the representative data, the average BMAL1 promoter expression level was 1.81 times higher in the Young control group than in the Old control group. The Vutiglabridin group showed 1.62 times higher average BMAL1 promoter expression level than the Old control group (Figure 4a).

BMAL1 promoter expression patterns of the Old control group showed dampened amplitude (Figure 4b). Compared to the Young control group, the first amplitude was dampened 0.52 times (*p* < 0.001), the second amplitude was dampened 0.91 times, and the third amplitude was dampened 0.88 times in the Old control group.

However, the BMAL1 promoter expression pattern of the Vutiglabridin group showed a greater increase in amplitude than the Old control group (Figure 4c). Compared to the Old control group, the first amplitude was increased 1.83 times in the Vutiglabridin group.

BMAL1 promoter expression patterns of the Old control group showed a significantly prolonged clock period compared to the Young control group as measured by the first period, second period, and cosinor period (Figure 4d–f): 24.85 ± 0.24 h Young control group vs. 27.22 ± 0.15 h Old control group; 25.20 ± 0.14 h Young control group vs. 26.03 ± 0.26 h Old control group; 24.97 ± 0.08 h Young control group vs. 25.92 ± 0.14 h Old control group.

However, the BMAL1 promoter expression pattern of the Vutiglabridin group showed a significantly shorter clock period than the Old control group. Each group’s representative detrended luciferase activity showed a significantly decreased period in the Vutiglabridin group (Figure 4c). In the representative data, the first trough of each group was similar (Vutiglabridin group: 23.66 h, 24.02 h; Old control group: 24.10 h, 23.62 h), but a notable difference was observed in the second trough (Vutiglabridin group: 50.66 h, 49.19 h; Old control group: 51.10 h, 50.62 h). The gap widened in the third trough time (Vutiglabridin group: 75.50 h, 74.18 h; Old control group: 79.93 h, 77.46 h). Shortened periods in the Vutiglabridin group were consistently observed as measured by first period, second period, and cosinor period (Figure 4d–f): 27.22 ± 0.15 h Old control group vs. 25.38 ± 0.29 h Vutiglabridin group; 26.03 ± 0.26 h Old control group vs. 25.16 ± 0.29 h Vutiglabridin group; 25.92 ± 0.14 h Old control group vs. 24.92 ± 0.04 h Vutiglabridin group.

Also, the circadian clock period of the Vutiglabridin group, as measured by first period, second period, and cosinor period, did not show a significant difference compared to the Young control group, suggesting strong normalization of circadian rhythmicity induced by vutiglabridin (Figure 4b–d): 24.85 ± 0.24 h Young control group vs. 25.38 ± 0.29 h Vutiglabridin group; 25.20 ± 0.14 h Young control group vs. 25.16 ± 0.29 h Vutiglabridin group; 24.97 ± 0.08 h Young control group vs. 24.92 ± 0.04 h Vutiglabridin group.

The above data illustrate the alleviation of senescent circadian phenotype in the Vutiglabridin group. Furthermore, dampened amplitude and prolonged periods with cellular senescence appeared significantly reduced in long-term vutiglabridin-treated HDFs.

## 4. Discussion

In this study, we found that vutiglabridin significantly ameliorates progression to cellular senescence and its associated dysfunctions in mitochondria and ROS generation and the circadian clock in primary HDFs.

The long-term culture of HDFs induced markers of cellular senescence in this study. One limitation was that the senescent cells in this study did not have complete cell cycle arrest, and vutiglabridin was administered during the progression to senescence rather than to fully senescent cells. This protocol was chosen in order to establish stable circadian rhythm experimentation, as fully senescent cells may not be resilient enough for the assay. Regardless, we found clear and robust signs of cellular senescence in the Old control group, as measured by cPDL, p16 and p21 gene expression, and the SA-β-gal assay, which is used to assess the progression of cellular senescence. Furthermore, vutiglabridin was not intended to be a senolytic drug, which selectively induces the death of senescent cells. The delaying or preventative effect of vutiglabridin on senescence progression was clearly observed in this study, suggesting that it has the potential to be developed as an anti-senescent drug.

Vutiglabridin was found to remarkably restore mitochondrial dysfunction, as evidenced by trends toward normalization of glycolysis, OCR, and morphology comparable to the Young control group. Vutiglabridin has been previously suggested to mediate changes in mitochondria [30], and its mechanistic effect on improving mitochondrial health was recently observed in the L-02 hepatocyte model, with impaired mitochondrial function induced by excess palmitate [31]; vutiglabridin was found to reduce mitochondrial ROS by enhancing the activity of a mitochondrial protein called paraoxonase-2, which is implicated in the maintenance of cellular ROS level. The connection between paraoxonase-2 [46] and cellular senescence has not been studied. The relationship between mitochondrial dysfunction and accelerated cellular aging, on the other hand, is gaining attention [47]. Various attempts have been made to target mitochondria to inhibit cellular senescence-associated secretory phenotype (SASP)—secretion of inflammatory cytokines and other senescence-related factors. Rapamycin was reported to reduce mitochondrial ROS and SASP [48], and metformin was illustrated to slow cellular senescence by affecting several signalling pathways, including mitochondrial oxidative phosphorylation [49]. In this study, treatment with vutiglabridin has revealed the amelioration of mitochondrial function, in addition to delaying the progression of cellular senescence, which further supports its potential development.

The effects of vutiglabridin on the metabolic regulators—AMPK and NAD^+^—were investigated in this study, and we found that vutiglabridin markedly trends to normalize their levels to the Young control group. AMPK phosphorylation is closely related to cellular senescence. In pulmonary emphysema, cellular senescence was reduced through increased phosphorylation of AMPK [50]. The aging-induced reduction in AMPK phosphorylation may also affect the circadian clock’s prolonged period and dampened amplitude. By increasing the phosphorylation of AMPK through metformin, the circadian clock period was reported to decrease in Rat-1 fibroblasts [51]. Vutiglabridin-treated HDF groups had a higher pAMPK level than the Old control groups, suggesting that the senescent circadian phenotype could have been alleviated through the metabolic regulation by vutiglabridin. In addition, the level of NAD^+^ was restored in HDFs through the vutiglabridin treatment. Declined NAD^+^ and NAD^+^/NADH ratios are well-known features of senescent cells [52], and a decreased NAD^+^ level affects the SIRT family and mitochondrial homeostasis. Many pharmacotherapy strategies aim to recover the level of NAD^+^ in organs to treat age-related metabolic diseases [53]. Here, we show that vutiglabridin as a clinical-stage compound (clinical phase 2 study in obese patients, NCT05197556) increases the NAD^+^ level and NAD^+^/NADH ratio and alleviates senescence-related metabolic changes in HDFs.

In addition to alleviating senescence-related metabolic changes, the senescent phenotype of the circadian clock was restored by the vutiglabridin treatment. Vutiglabridin-treated HDF groups showed increased BMAL1 promoter expression, widened amplitude, and shortened period, demonstrating normalization of cellular circadian rhythmicity. It might be postulated that alleviation of senescence-associated irregular circadian clock rhythmicity could be induced by metabolic regulation since vutiglabridin restored the reduced NAD^+^ and NAD^+^/NADH ratio and mitochondrial activity in senescent cells.

The circadian clock and metabolic reprogramming have long been studied as essential modulators of aging. A recent study that the small molecule nobiletin, which induces enhancement of the circadian rhythm, treats metabolic syndrome, and triggers healthy aging of skeletal muscle, suggests the possibility of using a clock enhancer for overcoming and treating aging [54]. Similarly, this study showed that at the cellular level, restoring the metabolic changes that occur during cellular senescence by a novel compound, vutiglabridin, delays cellular senescence and alleviates changes in cellular circadian rhythmicity. As a limitation, the causative roles between metabolic regulation and circadian rhythm were not examined in this study. Further study is needed to investigate the causal relationship between metabolic regulation, senescence, and circadian rhythm and whether relieving metabolic deregulation during aging at the organism level may also improve changes in the organism’s circadian rhythm. Of note, a transcription coactivator peroxisome proliferator-activated receptor gamma coactivator 1-alpha (PGC-1α) may be a possible link between mitochondrial and metabolic regulation, cellular senescence, and circadian rhythm. PGC-1α is a master regulator of mitochondrial biogenesis and has recently been reported to regulate mitochondrial network dynamics and mitophagy [55]. It has been reported to be rhythmically expressed in synchronized human hepatocytes consistent with BMAL1 expression [56] and to induce expression of various core clock genes, such as BMAL1, CLOCK, and PER2; [57] the relationship between PGC-1α and vutiglabridin may be examined in further studies.

Taken together, treatment with the anti-obesity drug candidate vutiglabridin to the senescent HDFs showed general alleviating effects on senescence-associated changes in metabolic reprogramming, mitochondrial dysfunction, and irregular circadian rhythmicity, which strongly implicate the overall adjusting effect of the compound on the progression of cellular senescence.

## 5. Conclusions

We found that vutiglabridin alleviates the progression of cellular senescence and associated dysfunctions in mitochondria and ROS generation and the circadian clock in primary HDFs. Chronic treatment with vutiglabridin changed the expression pattern of metabolic regulators, such as AMPK and NAD^+^. By alleviating senescence-related metabolic changes, the senescent phenotype of the circadian clock was restored.

These data propose the potential development of vutiglabridin as a drug of aging-associated diseases and aging-related metabolic diseases. This corroborates the intricate link between cellular senescence, metabolism, and the circadian clock.

## Figures and Tables

**Figure 1 antioxidants-13-00109-f001:**
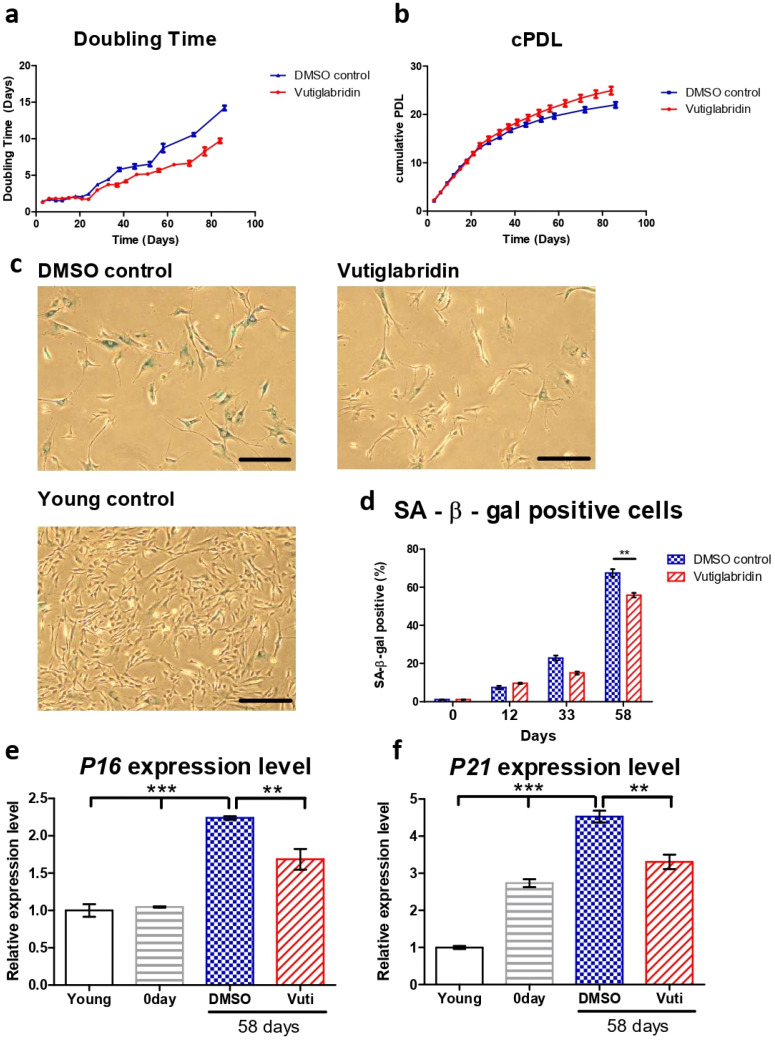
Vutiglabridin alleviates replicative senescence of HDFs. (**a**,**b**) Doubling time and cumulative population doubling level (cPDL) over time of HDFs cultured with 20 μM of vutiglabridin (Vutiglabridin) or vehicle (DMSO control), starting from passage 32 along with vutiglabridin treatment. HDFs were treated with 20 μM of vutiglabridin (Vutiglabridin) or an analog volume of DMSO as a vehicle (DMSO control). Data are shown as means ± SEM from three independent experiments (n = 3). (**c**) Photomicrograph of SA-β-gal staining of HDFs cultured with 20 μM of vutiglabridin (Vutiglabridin) or vehicle (DMSO control) for 58 days, starting from passage 32. The scale bar represents 100 μm. (**d**) Quantification of SA-β-gal staining. At least 200 cells per sample were counted as either positive or negative. Data are shown as means ± SEM from three independent experiments (n = 3). (**e**,**f**) The relative gene expression of P16 and P21 in HDFs in passage 14 (Young), HDFs in passage 32 (0 day), HDFs treated with 20 μM of vutiglabridin for 58 days from passage 32 (Vutiglabridin day 58), and HDFs treated with vehicle for 58 days from passage 32 (DMSO day 58) in comparison to the expression level in Young control is adjusted to 1. Data are shown as means ± SEM from three independent experiments (n = 3). ** *p* < 0.01, *** *p* < 0.001 by student’s two-tailed *t*-test.

**Figure 2 antioxidants-13-00109-f002:**
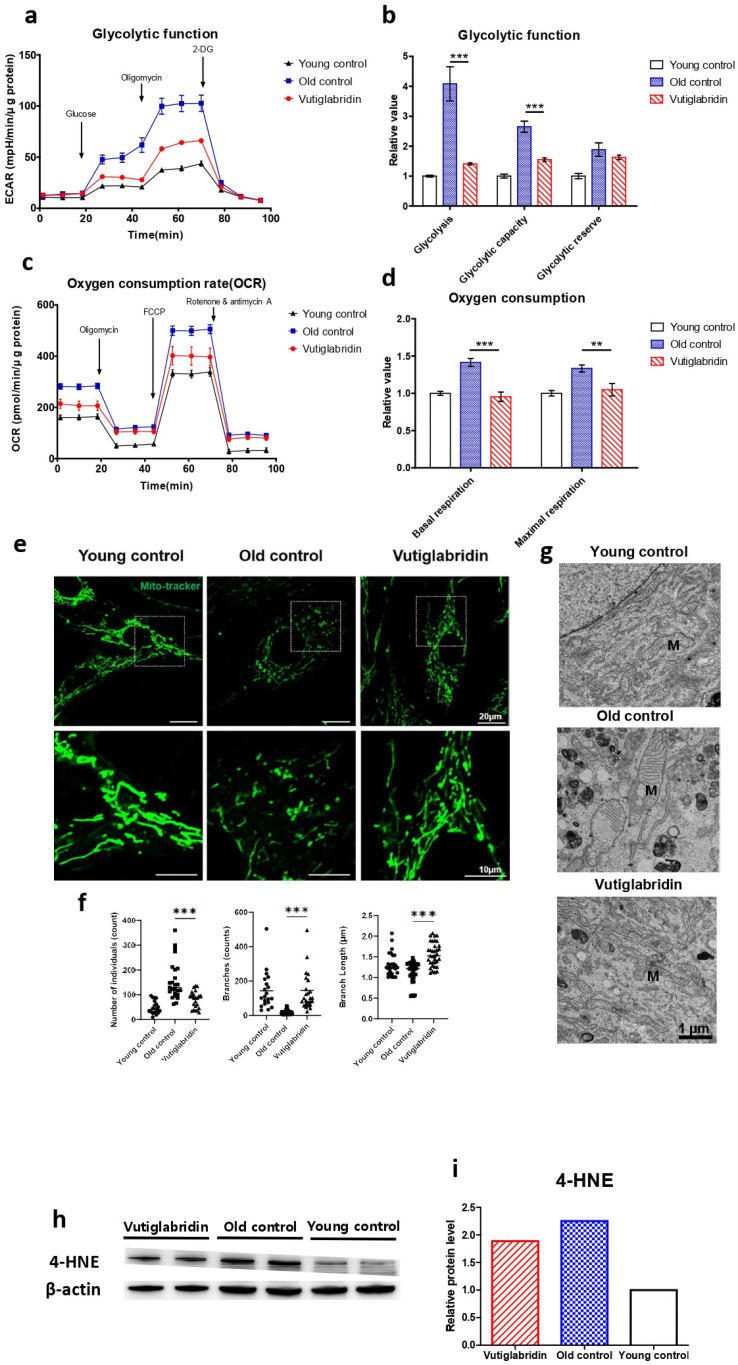
Mitochondrial dysfunction was reduced in the Vutiglabridin group. Vutiglabridin mitigates mitochondrial dysfunction and oxidative stress of aging HDFs. (**a**) A representative graph output from the XFe24 analyzer showing the ECAR response to glucose, oligomycin, and 2-deoxyglucose (2-DG). The glycolytic activities were assessed by measuring the ECAR with the Seahorse XFe24 analyzer. HDFs treated with 20 μM of vutiglabridin (Vutiglabridin) or an analog volume of vehicle (Old control) for 58 days from passage 32. HDFs in passage 14 were used as a young control (Young control). The ECAR values (mpH/min) were normalized to μg of protein. Each data point represents means ± SEM (n = 5). (**b**) Glycolytic activities: glycolysis, glycolytic capacity, and the glycolytic reserve of Old control, Vutiglabridin, and Young control. As described, relative values are generated in Figure 2a [35]. The glycolysis rate was calculated as the difference between the ECAR level before and after injecting glucose. Glycolytic capacity was calculated as the difference between the ECAR level before injecting glucose and after ejecting oligomycin. The glycolytic reserve was calculated as the difference between the ECAR level before and after ejecting oligomycin. Each data point represents means ± SEM (n = 5; *** *p* < 0.001 by student’s two-tailed *t*-test). (**c**) A representative graph output from the XFe24 analyzer showing the OCR response to oligomycin, fluoro-carbonyl cyanide phenylhydrazone (FCCP), and rotenone/antimycin A. HDFs treated with 20 μM of vutiglabridin (Vutiglabridin) or an analog volume of vehicle (Old control) for 58 days from passage 32. HDFs in passage 14 were used as a young control (Young control). All OCR values (pmol/min) were normalized to μg of protein. Each data point represents means ± SEM (n = 5). (**d**) Plots of basal respiration and maximal respiration of Old control, Vutiglabridin, and Young control. Data extracted from the analyses in Figure 2c. Relative values were generated from Figure 2c in accordance with Divakaruni et al. [36]. Basal respiration was calculated as the difference between the OCR level before injecting oligomycin and after injecting rotenone/antimycin A. Maximal respiration was calculated as the difference between the OCR level before and after injecting rotenone/antimycin A. Every value in Young control is normalized to 1. Each data point represents means ± SEM (n = 5; ** *p* < 0.01, *** *p* < 0.001 by student’s two-tailed *t*-test). (**e**) Morphological changes of mitochondria in light microscopy. Confocal imaging of cells treated with 200 nM Mitotracker showed mitochondrial structures. The scale bar is 20 µm. (**f**) The bars represent median values, and each black dot represents different data points (n = 20–30; *** *p* < 0.001 by student’s two-tailed *t*-test). (**g**) The ultrastructure of mitochondria in electron microscopy. M: mitochondria. The scale bar is 1 µm. (**h**) Total lysates from 20 μM of vutiglabridin or the same volume of DMSO-treated 58-day HDFs were subjected to western blotting using specific antibodies for 4-HNE and β-actin. (**i**) Mean value graphs for quantification of Figure 2h.

**Figure 3 antioxidants-13-00109-f003:**
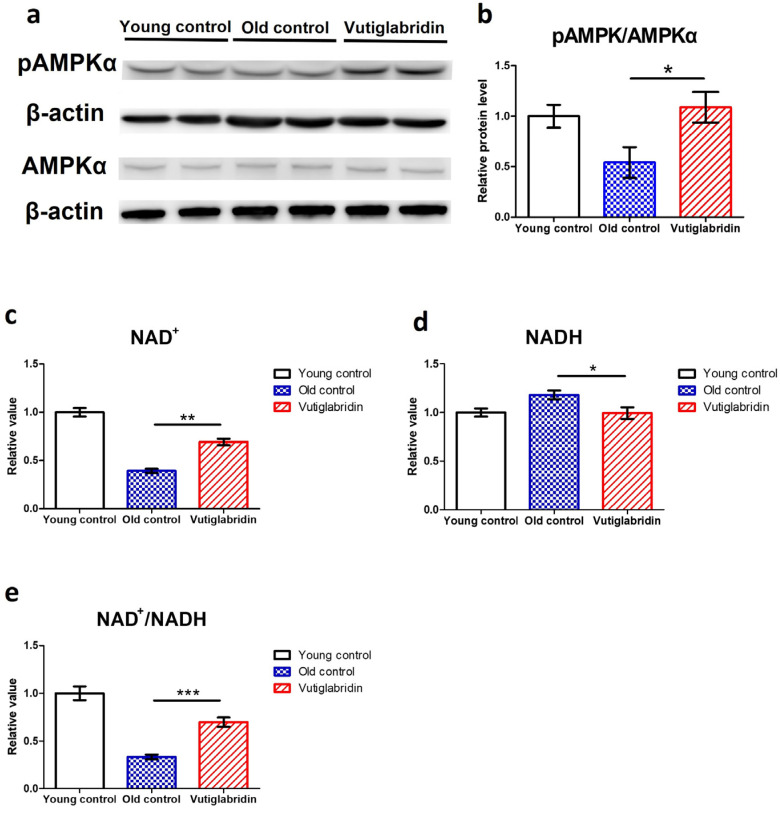
Vutiglabridin restoring the expression of the metabolic regulatory proteins in aging HDFs was changed in the Vutiglabridin group. (**a**) Total lysates from Young HDFs or 20 μM of vutiglabridin or the same volume of DMSO-treated 58-day HDFs were subjected to western blotting using specific antibodies for phosphorylated(p)-AMPKα, AMPKα, and β-actin. (**b**) pAMPKα/AMPKα ratios were quantified. Data are shown as means ± SEM (n = 4). (**c**–**e**) Concentrations of NAD^+^ (**c**), NADH (**d**), and the ratio of NAD^+^/NADH (**e**) were measured in HDF groups. Young control is passage 14 HDF groups. Old control is 58 days DMSO-treated HDF groups. Vutiglabridin is 58 days vutiglabridin-treated HDF groups. Values in the Young control were normalized to 1. Data are shown as means ± SEM (n = 3, * *p* < 0.05, ** *p* < 0.01, *** *p* < 0.001 by student’s two-tailed *t*-test).

**Figure 4 antioxidants-13-00109-f004:**
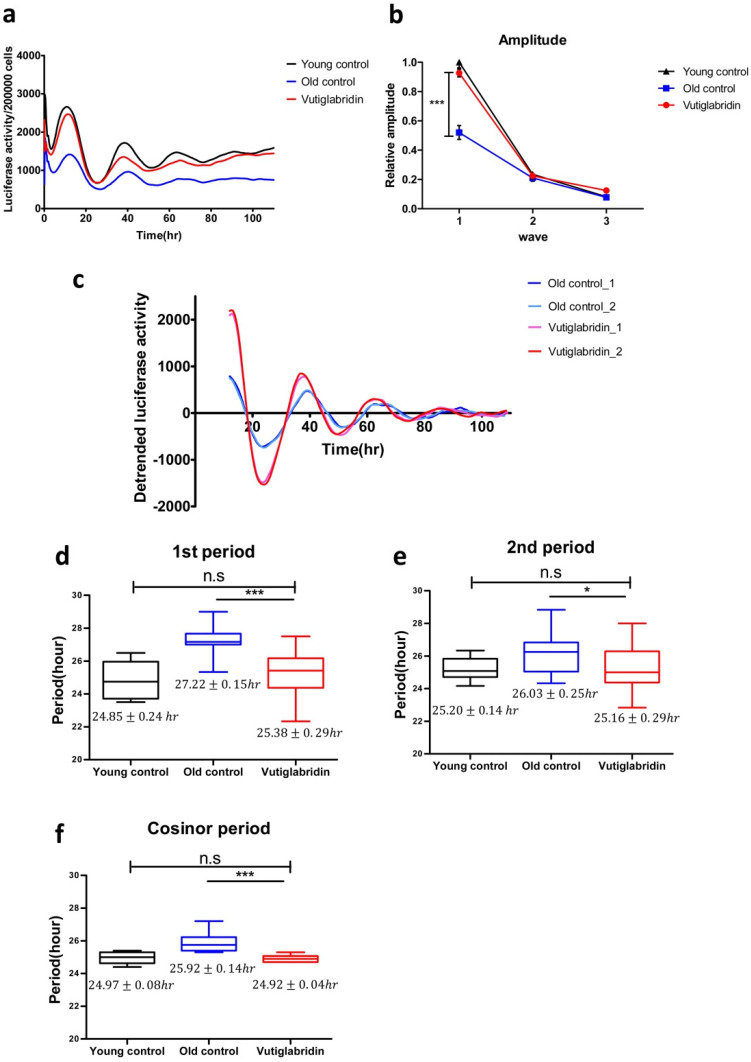
Vutiglabridin restores the circadian clock by shortening the period and increasing the amplitude of BMAL1 promoter expression in aging HDFs. (**a**) The effect of vutiglabridin on the rhythmic expression profile of the BMAL1 promoter linked to luciferase (Luc) was measured in HDFs for 5 days using Kronos-Dio (Atto). Representative luciferase activity in BMAL1^Luc^ HDFs measured from Kronos Dio Real-Time luminometer. Luciferase activity was normalized by cell counting at the endpoint. Young control represents BMAL1^Luc^ HDFs derived from passage 14 HDFs. Old control represents BMAL1^Luc^ HDFs derived from cells cultured with a medium with vector (DMSO) for 58 days. Vutiglabridin represents BMAL1^Luc^ HDFs derived from cells administered with vutiglabridin for 58 days. (**b**) The amplitude of luciferase activity graphs from stable BMAL1^Luc^ HDF groups was measured. 1st amplitude was defined as the difference of luciferase activity between the 1st peak and 1st through. 2nd and 3rd amplitudes were defined similarly to the first. Data are shown as means ± SEM (four independent experiments were performed, n = 9~10, *** *p* < 0.001, by student’s two-tailed *t*-test). (**c**) Representative detrended luciferase activity in BMAL1^Luc^ HDFs measured from Kronos Dio Real-Time luminometer. (**d**–**f**) Box-whisker plots of 1st period (**d**), 2nd period (**e**), and cosinor period (**f**). 1st period was defined as the interval between 1st through and 2nd through. 2nd period was defined as the interval between 2nd through and 3rd through. Cosinor analyzed period was measured mathematically using the cosinor software provided by Dr. R Refinetti (https://www.circadian.org/softwar.html, accessed on 7 November 2021). Data are shown as means ± SEM (eight independent experiments were performed, n = 20, ns., no signal. * *p* < 0.05, *** *p* < 0.001 by student’s two-tailed *t*-test).

## Data Availability

The underlying data of this manuscript are available from the corresponding authors upon reasonable request.

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
