# Peer review of "Vutiglabridin Alleviates Cellular Senescence with Metabolic Regulation and Circadian Clock in Human Dermal Fibroblasts"

_antioxidants, 2024, doi:10.3390/antiox13010109_

Round 1

Reviewer 1 Report

Comments and Suggestions for Authors

The authors report an in vitro study aimed at assessing the effects of vutiglabridin, a synthetic more stable analogue of the natural compound glabridin, on senescent parameters.

These data are interesting; however, it would have been more convincing to compare vutiglabridin with glabridin for these parameters, and in particular for those with a possible medical application.  Glabridin has been shown to exert various useful properties in medicine, however the senolysis is not one of the most important ones.

Author Response

Glabridin has various pharmacological effects, which are well proven in vivo. The categories of pharmacological effects for which glabridin has been proven include anti-inflammatory, anti-metabolic abnormalities, anti-cancer, and neuroprotection (1). Vutiglabridin has improved bioavailability based on the structure of glabridin and has proven to be more effective in treating obesity than glabridin (2). There are few studies that have directly proven the anti-aging effect of glabridin. The reason we conducted anti-aging research on vutiglabridin is because the existing pharmacological efficacy of glabridin is a phenomenon that appears during the aging process. As aging process, metabolic dysfunction including mitochondria dysfunction, proteostatic dysfunction and chronic inflammation occurs. We have already proven its anti-inflammatory and weight loss effects to be superior to glabridin (1, 3). Vutiglabridin was confirmed to have the effect of reducing mitochondrial ROS by promoting autophagy/mitophagy through PON2 protein (4). Based on these pharmacological effects, vutiglabridin produces an anti-aging effect by ameliorating metabolic dysfunction and inflammation.

<Reference>

  • Chun-xiao Li et al, Pharmacological properties of glabridin (a flavonoid extracted from licorice): A comprehensive review, Journal of Functional Foods, Vol 85, 2021, 104638,
  • Choi, L.S., Jo, I.G., Kang, K.S. et al. Discovery and preclinical efficacy of HSG4112, a synthetic structural analog of glabridin, for the treatment of obesity. Int J Obes 45, 130–142 (2021).
  • Shin J, Choi LS, et al. Synthetic Glabridin Derivatives Inhibit LPS-Induced Inflammation via MAPKs and NF-κB Pathways in RAW264.7 Macrophages. Molecules. 2023 Feb 24;28(5):2135.
  • Shin, G.-C., Lee, H. et al. (2021). Synthetic glabridin derivatives mitigate steatohepatitis in a diet-induced biopsy-confirmed non-alcoholic steatosis hepatitis mouse model through paraoxonase-2. bioRxiv, 462722,

Reviewer 2 Report

Comments and Suggestions for Authors

The manuscript by Heo and co-authors describes the effects of Vutiglabridin   on cellular senescence, cell metabolism and circadian clock proteins in human dermal fibroblast (HDF). The study provides novel information in the area of senescence research. In general, the paper is well written and addresses an interesting and important topic with convincing data. The experimental progression is logical and justified. In Introduction  obesity is described as an example of  imbalanced  metabolic homeostasis and cellular senescence.

Major point. The goal of the study was to examine the effects of Vutiglabridin, a chemically modified glabridin, an  anti-obesity  compound,  on metabolic changes associated with senescence and circadian clock disfunction.  In this regard it would be more logical to study the circadian clock  and metabolic effects of Vutiglabridin not only in HDF but also in adipocytes, which are also prone to senescence.

One more  point: Fig.1 e,f depicts the graphs of p16 and p21 expression. It would be important  to see the original western blots of these proteins.

Author Response

Major point. The goal of the study was to examine the effects of Vutiglabridin, a chemically modified glabridin, an anti-obesity compound, on metabolic changes associated with senescence and circadian clock dysfunction.  In this regard it would be more logical to study the circadian clock and metabolic effects of Vutiglabridin not only in HDF but also in adipocytes, which are also prone to senescence.

--Human dermal fibroblasts were used because the primary focus of the experiments in this manuscript was to examine whether anti-obesity drugs could have extended efficacy against age-related degenerative diseases. However, we fully agree that adipocytes would be highly suitable for cellular investigations aimed at elucidating the correlation between ageing and metabolism. Currently, our team is engaged in a comparable experiment using hepatocytes, and we will also incorporate adipocytes into it. However, this would require long-term experimental planning and validation, so it would be difficult to add it to the manuscript we are submitting now.

One more point: Fig.1 e,f depicts the graphs of p16 and p21 expression. It would be important to see the original western blots of these proteins.

--We agree that adding western blots data to Fig1 e, f would make the data more convincing. However, due to the nature of the experiment, which requires sampling after inducing replicative senescence, it will take at least 4-5 weeks of additional time. Therefore, it would be difficult to revise the manuscript with relevant western blot data within 10 days. Also, we would appreciate it if you could consider that we are providing data from several experiments(cPDL, SA-beta-gal) in addition to p16 and p21 data to demonstrate cellular senescence in Fig.1.

Reviewer 3 Report

Comments and Suggestions for Authors

The manuscript presents interesting data concerning the effect of vutiglabridin, a chemically modified derivative of glabridin, on the senescent human dermal fibroblasts. The study revealed that chronic treatment of senescent primary human dermal fibroblasts (HDFs) with vutiglabridin alleviated all investigated markers of cellular senescence (SA- -gal, CDKN1A, CDKN2A) and dysfunctional cellular circadian rhythm (BMAL1), while preventing the alterations of mitochondrial function and structure occurring during cellular senescence, demonstrate the significant senescence-alleviating effects of vutiglabridin, specifically with the restoration of cellular circadian rhythmicity and metabolic regulation. In fact, as admitted in the Discussion, the study was done on cells in progression to senescence rather than fully senescent cells but it does not affect the validity of conclusions.

Remarks:

Title: The abbreviation should not appear in the title as it is not obvious to everybody

Line 52: “regulars”, I guess “regulators”

Line 128: “8*104 cell”, please, “4” in superscript

Lines 131/132: How 20 uM was “optimized as the human-relevant dose’? The concentration seems quite high for in vivo

Line 182 and other: “2-DG”, please explain acronyms on the first use

Line 209: what is “10% gradient Tris-glycine gel”?

Results: It is unusual to start this section with a Figure. Usually some text is first, and the Figure appears after citation in the text.

Comments on the Quality of English Language

English is generally fine, with a few spelling errors

Author Response

Remarks:

Title: The abbreviation should not appear in the title as it is not obvious to everybody

-- Edited as requested.

Line 52: “regulars”, I guess “regulators”

-- Edited as requested (line 52, 2 page).

Line 128: “8*104 cell”, please, “4” in superscript

-- Edited as requested (line 128, 3 page).

Lines 131/132: How 20 uM was “optimized as the human-relevant dose’? The concentration seems quite high for in vivo

--This concentration has been used as the optimized in vitro dose that is clinically relevant. Glaceum has identified through preclinical and clinical trials that the plasma exposure of vutiglabridin (HSG4112) at 2 μg/ml concentration yields maximum efficacy. The steady-state pharmacokinetic data of vutiglabridin at 720 mg daily dose is shown below for reference. Given the molecular weight of vutiglabridin (354.44 g/mol) and the ratio between drug exposure in plasma and skin at its Tmax (timepoint at which maximum exposure is achieved, which is 6 h), which we found to be 1 to 3.34 based on the drug distribution study performed in standard Sprague-Dawley rat (table shown below), we have calculated that 18.8 μM is the clinically-relevant dose for this study. Therefore, 20 μM was used and concentration-response curve was deemed not necessary. This data is unfortunately out of the manuscript’s scope and can only be shared here.

Line 182 and other: “2-DG”, please explain acronyms on the first use

-- Edited as requested (line 185, 4 page).

Line 209: what is “10% gradient Tris-glycine gel”?

-- Edited as requested (line 212, 5 page).

Results: It is unusual to start this section with a Figure. Usually some text is first, and the Figure appears after citation in the text.

-- Edited as requested (7, 8 page).

Reviewer 4 Report

Comments and Suggestions for Authors

Comment 1. In the abstract, there is a typographic error “SA- -gal” instead of “SA-β-gal”.

Comment 2.  In the WB, the authors should add a densitometry quantification in the figure 2h, and they should improve the image of the WB because every samples should be run in the same gel.

Author Response

Reviewer4:

Comment 1. In the abstract, there is a typographic error “SA- -gal” instead of “SA-β-gal”.

-- Edited as requested (line 26, 1 page).

Comment 2.  In the WB, the authors should add a densitometry quantification in the figure 2h, and they should improve the image of the WB because every samples should be run in the same gel.

-- Edited as requested (11 page, Figure 2i and Figure legend were added).

Round 2

Reviewer 2 Report

Comments and Suggestions for Authors

the authors provide sufficient explanation about the limitation of the study